# Effectiveness of Transition Care Intervention Targeted to High-Risk Patients to Reduce Readmissions: Study Protocol for the TARGET-READ Multicenter Randomized-Controlled Trial

**DOI:** 10.3390/healthcare11060886

**Published:** 2023-03-18

**Authors:** Alexandre Gouveia, Marco Mancinetti, Daniel Genné, Marie Méan, Gregor John, Lukas Bütikofer, Drahomir Aujesky, Jeffrey L. Schnipper, Jacques Donzé

**Affiliations:** 1Department of Ambulatory Care, Center for Primary Care and Public Health (Unisanté), University of Lausanne, Rue du Bugnon 44, 1011 Lausanne, Switzerland; 2Department of Internal Medicine, HFR-Fribourg Cantonal Hospital, Chem. des Pensionnats 2-6, 1752 Villars-sur-Glâne, Switzerland; 3Department of Internal Medicine, Bienne Hospital Center, Chante-Merle 84, 2501 Bienne, Switzerland; 4Division of Internal Medicine, Lausanne University Hospital (CHUV), Rue du Bugnon 46, 1011 Lausanne, Switzerland; 5Department of Internal Medicine, Neuchâtel Hospital Network, Rue de la Maladière 45, 2000 Neuchâtel, Switzerland; 6Department of Internal Medicine, Geneva University Hospitals (HUG), Geneva University, Rue Gabrielle-Perret-Gentil 4, 1205 Geneva, Switzerland; 7Clinical Trials Unit Bern, Faculty of Medicine, University of Bern, Mittelstrasse 43, 3012 Bern, Switzerland; 8Department of Internal Medicine, Bern University Hospital, University of Bern, Freiburgstrasse 18, 3010 Bern, Switzerland; 9Division of General Internal Medicine, Brigham and Women’s Hospital, Harvard Medical School, 1620 Tremont St. BC-3, Boston, MA 02120, USA

**Keywords:** transitional care, transitions of care, hospital readmission, hospital admission, hospital discharge, coordination of care, patient education, medication reconciliation

## Abstract

Hospital readmissions within 30 days represent a burden for the patients and the entire health care system. Improving the care around hospital discharge period could decrease the risk of avoidable readmissions. We describe the methods of a trial that aims to evaluate the effect of a structured multimodal transitional care intervention targeted to higher-risk medical patients on 30-day unplanned readmissions and death. The TARGET-READ study is an investigator-initiated, pragmatic single-blinded randomized multicenter controlled trial with two parallel groups. We include all adult patients at risk of hospital readmission based on a simplified HOSPITAL score of ≥4 who are discharged home or nursing home after a hospital stay of one day or more in the department of medicine of the four participating hospitals. The patients randomized to the intervention group will receive a pre-discharge intervention by a study nurse with patient education, medication reconciliation, and follow-up appointment with their referring physician. They will receive short follow-up phone calls at 3 and 14 days after discharge to ensure medication adherence and follow-up by the ambulatory care physician. A blind study nurse will collect outcomes at 1 month by phone call interview. The control group will receive usual care. The TARGET-READ study aims to increase the knowledge about the efficacy of a bundled intervention aimed at reducing 30-day hospital readmission or death in higher-risk medical patients.

## 1. Introduction

Transition of care between hospital and ambulatory care represents high-risk period for patients [1]. Many complications can occur as patients are discharged from hospitals to the ambulatory setting, which may lead to unnecessary hospital readmissions [2,3]. Each new hospital admission is itself associated with a risk of new complications [4,5]. For these reasons, readmissions were of high interest in health policies in several countries and financial incentives were implemented to promote a reduction in hospital readmissions and increase quality in the transition of care [6].

It was shown that approximately 50% of these hospital readmissions were potentially preventable, and 30% of the readmissions were truly preventable [7,8]. To avoid these preventable readmissions, some transitional care interventions showed some promising results [9,10,11,12,13,14]. These include telephone follow-up, medication reconciliation, patient education, home visits, among others. A systematic review and meta-analysis published in 2014 showed that interventions were overall somewhat effective in reducing readmission [15]. However, interventions that were effective were the most complex to implement and the most resource intensive. Furthermore, most of these interventions’ studies were performed on specific patient populations such as patients with heart failure or chronic lung disease only, which is not transposable to most medical patients who frequently have multiple chronic conditions. Among studies that included general medical patients, half targeted elderly patients only. We know, however, that age is not a good predictor for readmission, and therefore, by only specifically targeting the older patients’ interventions, the group of patients with an actual high-risk of readmission may be overlooked [13,14]. Although all patients deserve a high-quality discharge process, more complex and costly interventions, such as post-discharge phone calls or home-based transitions coaching, should be targeted to the patients who are most likely to benefit. Therefore, efficiently improving transitions in care requires hospitals to target discharge interventions at those patients at higher risk of potentially avoidable readmission.

### 1.1. Current Evidence from Randomized Trials

A meta-analysis of 42 trials found a pooled relative risk of 0.82 in reducing hospital readmission [15]. These findings, however, do not help to answer the question, “which intervention to which patient?”, due to the heterogeneity of the interventions tested in various patient population. Very few trials aimed at targeting high-risk patients in a general medical patient population. In 2014, Dhalla et al. targeted high-risk patients in a randomized controlled trial using the LACE index [16]. However, in this study, about 90% of the patients were identified as high risk, which questions the usefulness or validity of the score used and may have contributed to the negative findings. In another randomized controlled trial, Balaban et al. targeted patients at high-risk for readmission, which were identified with at least one from five risk factors for hospital readmission, chosen from the literature and local experience [17]. The readmission rates did not differ between intervention and control patients.

### 1.2. Identifying Patients at High Risk for Readmission

It was already shown that clinicians are not accurately identifying which patients are at higher risk for readmission [18,19,20]. A systematic review in 2011 carried out by Kansagara et al. highlighted the lack of performance and validity of the existing prediction models in readmission [21].

Since then, the HOSPITAL score was widely validated in nearly 200,000 patients, across 16 hospitals and five countries. It is now, therefore, one of the most valid and easy-to-use prediction models to identify high-risk patients for hospital readmission (Table 1) [22]. One of the largest validation studies was performed in more than 117,000 patients in four different countries, and showed good performance of the HOSPITAL score, with a C-statistic of 0.72 [23]. A simplified version of the HOSPITAL score, in which the number of procedures was left out, and in which the criteria of “discharge from an oncology unit” was replaced by “discharge from an oncology unit or diagnosis of active cancer” and showed similar performance [24,25,26]. We selected the simplified HOSPITAL score to identify the high-risk patients (i.e., total score of six points or more) due to its wide validation, good performance, and simplicity.

### 1.3. Research Questions

Currently, we lack evidence of effectiveness regarding the impact of interventions in reducing hospital readmissions in general medical patients. We also lack evidence of efficiency in targeting true higher-risk patients for readmission. Therefore, our primary research aim is to assess the effectiveness of a standardized multimodal nurse-delivered transition care intervention compared to usual hospital care for patients hospitalized in medical departments who are at higher-risk of hospital readmission. The risk will be calculated through the simplified HOSPITAL score. The primary outcome will be a composite endpoint of 30-day unplanned readmissions and death. Secondary outcomes include the time to first 30-day unplanned readmission or death, patient’s perspective on quality of transition of care, main causes of readmission, and health care use within 30 days after discharge and costs of readmission.

## 2. Materials and Methods

### 2.1. Design

The TARGET-READ study is a multicenter single-blinded randomized controlled trial assessing the effect of a multimodal nurse-driven intervention to reduce 30-day hospital readmission in higher-risk medical patients calculated through the HOSPITAL simplified score. This intervention involves specific patient information about his/her diseases, general patient education about lifestyle, assessment of dependence level, medication reconciliation, follow-up care planning, and assessment of possible barriers of patient’s ability to carry out the discharge plan. The intervention group will be compared to the control group that receives usual care (no standardized education or medication review, nor phone call follow-up). The intervention group will have outcomes collected at day 3 and day 7, and both groups will have their outcomes collected after 30 days. The follow-up period of the trial will be up to 45 days after hospital discharge. A study overview is available in Figure 1. The respective ethics committees of institutions participating in this study approved this trial.

### 2.2. Setting and Recruitment

Adult inpatients planned to be discharged home from the medical department of one of the 4 participating hospitals (Hospital Center of Bienne, HFR-Fribourg Cantonal Hospital, Lausanne University Hospital, and Neuchâtel Hospital Network) will be screened for eligibility by the local study nurse. Screening logs will be kept at each site, and basic coded information (gender, age, risk of readmission based on simplified HOSPITAL score, etc.) regarding each screened patient will be locally collected. No compensation nor payment will be made to the participants.

Only patients at high risk for readmission will be invited to participate. The determination of the risk of readmission will be based on the simplified HOSPITAL score [24]. This score requires the collection of six variables (Table 1), which are part of data collected in usual care and available in the electronic health record. High-risk patients are those that obtain 6 points or more on the simplified HOSPITAL score.

Patients will be included if they meet all the following criteria:(a)Adult patients planned to be discharged home or nursing home;(b)Discharged from a medical department of one of the four participating centers;(c)Hospital stay of at least 24 h;(d)Patient at higher risk of 30-day readmission based on the simplified HOSPITAL score.

Patients will be excluded according to the following criteria:(a)Previous enrolment into the current study;(b)Not living in the country in the next 30 days;(c)No phone to be reached at;(d)Not speaking French or German (depending on the site);(e)Refusal to participate or unable to give consent (including cognitive troubles defined as a disorientation in time and space).

### 2.3. Randomization Procedure

All medical patients will be screened for eligibility as soon as their hospital discharge is planned (Figure 1). After a baseline evaluation for all inclusion and exclusion criteria (Table 2), each enrolled participant of all sites will be randomized centrally within the electronic data capturing system to the intervention or to usual care group using a computer-generated randomization list (allocation ratio 1:1) stratified by discharge site and by readmission risk category according to the simplified HOSPITAL score (intermediate vs. high risk, i.e., a simplified HOSPITAL score of 4 or 5 vs. ≥6), using a permuted block design with randomly varying block sizes of 2, 4, and 6. Given the nature of the intervention, it will be not possible to blind patients and study nurses. However, treatment allocation in the database will be coded (1/2) and the study nurses collecting the outcomes or working on data cleaning will be blinded to the group allocation. Patients will be specifically asked not to mention their group allocation in the final assessment.

### 2.4. Transition Care Intervention

The intervention will take place during the initial hospital admission (index hospitalization). The TARGET-READ intervention has two components that are performed by the study nurse: one during the hospitalization before the hospital discharge (pre-discharge component), and one after the hospital discharge (post-discharge component) (Table 3).

The pre-discharge component of the TARGET-READ intervention consists of the six following steps during the index hospitalization:Patient information about his/her diseases;General patient education about lifestyle;Dependence level assessment;Medication reconciliation;Follow-up care planning;Assessment of possible barriers of patient’s ability to carry out the discharge plan.

Documents will be handed only to participants in the intervention group at discharge (Table 4). All the documents are provided as Appendix A.

The post-discharge component of the TARGET intervention consists of the five following steps that are carried out at day 3 (D3) and 14 (D14) after inclusion and after the index hospitalization:Review of the health conditions, with health care advice, if needed;Verification of the primary care physician follow-up;Review the current list of medications;Patient education;Health care advice.

Every study nurse in the different recruiting centers will receive the same training for the TARGET intervention. The training will be led by the sponsor, individually and in-group, as well as theoretically and practically. Participants randomized in the control group will receive information and organization of the post-discharge care in accordance with local usual care.

### 2.5. Data Collection and Outcomes

Data will be collected at five time points: on screening (before inclusion), on visit 1 (inclusion, day 0), on visit 2 (day 3 ± 1), visit 3 (day 14 ± 1), and visit 4 (day 30 ± 15) (Table 3). Study nurses will collect data through direct interviews (screening and visit 1) or phone-call interviews (visit 2, 3 and 4). Readmission costs will be obtained locally at each recruiting hospital. Contact information including name, address, and phone number of the patient will be necessary to contact the patient for the two follow-up phone calls that are part of the intervention, and to collect the outcomes at 30 days for both groups. A study patient can withdraw his/her consent to participate in the trial at any time and without any declaration of reason. Data collected until the time point of withdrawal will remain in the database and will be used for analysis. Missing data may occur due to dropouts or deaths. The former will lead to the absence of all outcome information, as outcomes are only assessed once and will be accounted for by multiple imputation based on all available baseline information. The latter will not lead to missing data for the primary outcome, 30-day deaths, and time to primary outcome. For 30-day unplanned readmission, death is a competing event and the cumulative incidence at 30 days will be used. Count outcomes will be handled by including the observation time as offset in the Poisson regression. For patient satisfaction and medication adherence, multiple imputations will be used. In a sensitivity analysis, the worst possible outcome will be assigned to deaths (i.e., a negative respond to all question of the CTM-3).

### 2.6. Primary Outcome Measurements

The primary outcome will be the composite of first unplanned readmission or death within 30 days after discharge of index admission. The unplanned readmission will be defined as a non-elective hospitalization that occurs within 30 days after discharge from the index hospital discharge to any division of any acute care hospital. Elective hospitalization will be defined as a non-urgent hospitalization scheduled at least 1 day prior to the admission day. Death will be defined as any death occurring within 30 days after discharge. We will further collect the cause of readmission or death. Death is the worst outcome after a hospital discharge. It is common practice in this domain to include both readmission and death as a composite outcome because patients who died may well have been readmitted if they had not died [20,27,28,29].

A blinded study nurse outside the participating site will collect the primary outcome at least 30 days after inclusion via phone interview. The blinded study nurse will only have access to the contact information from an Excel file and the outcome collection form in a browser-based research electronic data capture software (REDCap) to remain unaware of the group allocation of the participants. At time of inclusion, the participants will be asked to not divulgate their group allocation during the phone interview. In order to collect the primary outcome and reduce missing outcomes thoroughly, the outcomes will be assessed through five steps:(a)Phone interview with the study participant (minimum three phone calls attempts on at least two different days);(b)If study participant is not reachable, phone interview with a next of kin of the study participant (minimum two phone calls attempts on at least two different days);(c)If study participant and next of kin are not reachable: phone interview with the primary care physician of the study participant;(d)If none of the calls above allows the outcome collection, the nurse will call the study nurse of the local site to look for any outcome event that would be reported in the electronic health system at the inclusion site;(e)For all readmissions or deaths: the medical report will be collected to assess the exact place, time, length of stay, and cause of hospital readmission or death.

### 2.7. Secondary Outcome Measurements

Secondary outcomes are the following:Individual components of the primary composite outcome: 30-day unplanned readmission, or 30-day mortality;Time to first 30-day unplanned readmission or death;Main cause of readmission or death;Post-discharge health care utilization within 30 days after discharge from index hospitalization: number of unplanned and planned hospital readmission(s), total number of planned and unplanned days of hospitalizations, number of emergency room visits and primary care provider visits;Patient’s perspective (satisfaction) on quality of transition of care between hospital and home assessed by the three-item care transition measure (CTM-3) [30] at 30 days;Costs of readmission.

The 3-item care transitions measure (CTM-3) is a three-question survey that is a hospital level measure of the quality and effectiveness of the transition or discharge from an acute care hospital [30]. The CTM-3 measures whether patients’ preferences were followed during their care (patient centered care), and measures patients understanding of their own role in their care and their discharge medications. The three items are:(1)The hospital staff took my preferences and those of my family or caregiver into account in deciding what my health care needs would be when I left the hospital;(2)When I left the hospital, I had a good understanding of the things I was responsible for in managing my health;(3)When I left the hospital, I clearly understood the purpose for taking each of my medications.

We will measure the proportion of patients who respond positively to all three items 30 days after discharge of index admission.

The secondary outcomes will be collected by phone interview at the same time of the primary outcome and with the same procedure:(a)Number of unplanned readmissions: if the patient, the next of kin, or the primary physician report a readmission, the date of readmission and the length of stay will be asked. The medical report of the readmission will be systematically collected to double check the validity of the information and obtain the causes of readmission;(b)Number of days of hospitalization within 30 days: if the patient, the next of kin, or the primary physician report one or more readmission, the total number of days spent at the hospital will be measured through the electronic health system for the participating site, or by calling the hospital if a readmission occurred outside the four participating sites;(c)Diagnoses at readmission or cause of death: if the patient, the next of kin, or the primary physician report a readmission, the medical report will be collected. If a death is identified, the medical report will be collected if the death occurred at the hospital, otherwise the cause will be obtained from the primary care physician, or the certificate of death;(d)Number of emergency visits within 30 days: the patient, the next of kin, or the primary physician will be asked about the number of visits to any emergency room (any clinic or hospital). Number of primary care physician visits: the patient, the next of kin, or the primary physician will be asked about the number of visits to any treating physician (primary care or specialist);(e)Patient’s perspective (satisfaction) on quality of transition of care between hospital and home: only the patient (or a caring person) can answer the questionnaire 3-item care transition measure (CTM-3);(f)Costs of readmission: the overall costs of the readmission will be collected at the hospital site.

### 2.8. Data Analysis

The main analysis set will include all randomized patients according to the intention-to-treat principle. A secondary per protocol analysis set will include patients who received the pre-discharge component plus at least one of the two post-discharge follow-up phone calls of the intervention, did not violate inclusion or exclusion criteria, were discharged home or nursing home, did not cross over (i.e., control patients receiving any part of the intervention), and did have the same HOSPITAL score risk group at randomization and discharge.

The primary outcome will be reported as a proportion of patients in each group with a 95% Wilson score confidence interval (CI) and compared between the groups by a Mantel–Haenszel risk difference stratified for the stratification factors used in randomization (i.e., discharge site and readmission risk category). A two-sided 95% CI will be calculated according to the procedure described by Klingenberg [31]. A stratified Cochran–Mantel–Haenszel test will be used to test for differences. Thirty-day deaths and dichotomized patient satisfaction will be analyzed in the same way. The proportion of 30-day unplanned readmissions in each group will be estimated using the cumulative incidence function with death as competing event calculated from flexible parametric survival models [32] and compared by a risk difference with 95% CI. We will also report the risk difference of the competing event (death without readmission) at 30 days for each group and the difference between groups. Time to unplanned readmission or death will be compared between groups using a log-rank test stratified for the stratification factors used in randomization. As an effect measure, we will use the restricted mean survival time truncated at 30 days calculated by flexible parametric survival models with the group and factors used at randomization as covariates [33]. The restricted mean survival time for each group and the difference between groups will be reported with 95% CI and a *p*-value. Count outcomes (number of hospital readmissions, number of days of hospitalization, number of emergency room visits, number of primary care provider visits) will be presented with number of patients, person-time, and incidence rate with 95% CI. Groups will be compared using a negative binomial regression with the group and the stratification factors as covariates and the observation time as offset. An incidence rate ratio with 95% CI and *p*-value will be reported. In case of an excess of zeros and over dispersion that cannot be modeled by the negative binomial distribution, we will consider zero-inflated negative binomial regression. Each item of the CTM-3 score will be summarized by treatment group using relative and absolute frequencies. The number of patients with a yes on all items will be compared between treatment groups using a Mantel–Haenszel risk difference with a two-sided Klingenberg 95% CI and a Cochran–Mantel–Haenszel test, both stratified for the stratification factors used in randomization. The costs of readmission will only be analyzed for patients that had a readmission using linear regression with the treatment group and the stratification factors used in randomization as covariates. The treatment effect will be presented as mean difference with 95% CI and a *p*-value. Model assumption will be determined visually using plots of residuals (residuals vs. fitted values, QQ-plot). If model assumptions are violated, transformation of the outcome (e.g., log), more robust methods (e.g., robust standard errors or robust regression), or non-parametric methods (e.g., Wilcoxon-Mann–Whitney test or van Elteren’s test) will be considered. For all outcomes, a secondary analysis will be carried out on the per-protocol set.

In a sensitivity analysis for the primary outcomes, all patients with readmissions within the first 24 h will be excluded. In further sensitivity analyses, we will analyze all outcomes without adjusting for stratification and estimate the primary outcome based on survival methods. The primary outcome will be analyzed in subgroups defined by risk for readmission (HOSPITAL score of 4 and 5 vs. ≥6), clinical site, whether the patient suffers from diabetes, chronic heart failure, COPD, or cancer, living place (nursing home vs. rest), living status (alone vs. rest), and health insurance (private vs. rest). Subgroups will be analyzed using regression models with the treatment group, the subgroup and their interaction and the stratification factors used at randomization as covariates. If possible, a binomial model with identity link function will be used and the effects will be reported as risk differences with 95% CIs. Otherwise, a Poisson model with identity link and robust standard errors (leading to risk differences) or a binomial model with logit link (leading to odds ratios) will be considered. Models with and without interaction will be compared using a likelihood ratio test and the *p*-value will be reported as *p*-value for interaction. Results will be presented in a forest plot.

Since the follow-up is very short, we do not expect many dropouts and it is unlikely that these patients will have a readmission or will die. Therefore, we will assume that dropouts will not have a readmission and will not die. For survival analyses, they will be censored after one hour. For count outcomes, they will be assumed to have no observed event and an offset of one hour. For patient satisfaction, multiple imputations will be used. If the number of dropouts is larger than 5%, we will conduct a sensitivity analysis in which the primary outcome (unplanned readmission or death) will be multiply imputed as a binary variable. For death or readmission at the day of discharge, we will use an event time of half a day. For unplanned readmission or death at an unknown date, we will use the median time to unplanned readmission or death of all patients with the event in the corresponding treatment group. The cost of readmission will only be analyzed for patients with a documented readmission. Missing data will be multiply imputed in a separate model with only those patients.

### 2.9. Interim Analysis

No interim analysis is planned.

### 2.10. Sample Size Calculation

The intervention phase will be restricted to the eligible patients who are at intermediate to high risk of 30-day readmission according to the simplified HOSPITAL score, i.e., ≥4/13 points (Table 1). Around 18,000 patients are planned to be discharged from all participating sites during the 20-month study period, 30% of which (5400) are estimated to be at higher-risk for a 30-day readmission or death (Figure 2). Because we will target patients at higher risk of readmission, we hypothesize that the intervention could reduce readmission by 25%, i.e., more than the 18% reduction found in a recent meta-analysis where patients were mostly not at high risk for readmission [15]. Based on previous findings, the expected 30-day readmission and death rate for patients at intermediate or high risk according to the simplified HOSPITAL score is around 27%. Allowing for 10% loss to follow-up, we determine that we will need 1380 patients (690 in each arm) for the study to have 80% power at an alpha of 5%, based on a chi-squared test. According to a conservative estimation, the sample size is expected to be reached within the study period.

### 2.11. Trial Status

This trial has finished enrolling participants. Data collection began on 3 April 2018 and finished on 31 December 2019. The study was registered in ClinicalTrials.gov (Registration Number: NCT03496896; https://clinicaltrials.gov/ct2/show/NCT03496896, accessed on 22 February 2023).

## 3. Discussion

We presented the TARGET-READ study protocol, which evaluates the effectiveness of a transition care intervention delivered by nurses to medical high-risk patients planned to be discharged from four Swiss hospitals in reducing 30-day unplanned hospital readmissions or death. We will compare the effectiveness of this intervention to usual care.

Although research carried out in this field was able to prove the effectiveness of complex interventions in reducing hospital readmissions, there are gaps in knowing if high-risk medical patients, identified by the internationally validated simplified HOSPITAL score, benefit or not from a bundled intervention delivered by a nurse during the transition period. This study will also provide relevant information regarding the use of health care after discharge for these high-risk patients, as well as their satisfaction in the way that care during transition was delivered.

This study has some strengths. The effectiveness of transitional care intervention to reduce readmission showed inconsistency, and a new large RCT may add valuable results. Very few studies investigated a set of intervention to reduce readmission that can be used in a general medical population independently from their main diagnosis, and we will include in this study all medical patients, regardless of the cause of admission and with multiple chronic conditions as in real-life, which might provide more useful and generalizable results. We will target the higher risk patients using one of the most accurate and widely validated tools, the HOSPITAL score, to better allocate transition care resources and to improve effectiveness, which is a step further from most of the research in the field. We will include four hospitals including mid-size teaching hospitals and a tertiary care hospital. We will perform the first randomized controlled trial to test a transition care intervention in medical patients in Switzerland. The satisfaction of the transition of care will be evaluated.

We consider, however, that our study has some limitations. Firstly, we are aware that recall bias might be relevant since we will obtain information through telephone calls to patients and their caretakers. Local study nurses will also try to collect relevant information for primary and secondary outcomes through local medical records from the discharge hospital and ambulatory healthcare providers, to minimize recall bias. Secondly, patients with mental impairment are excluded from our study, which limits generalizability, and close relatives are not invited to participate to avoid increasing heterogeneity in the intervention group. Thirdly, this multicentric study will be conducted in hospitals with different profiles, some more regional and peripheral, while others urban and academic, as well as in different languages (French or German-speaking cantons). This might be relevant to the patients included in our study in the different centers, as well as the available ambulatory care resources in different areas.

## 4. Conclusions

This RCT aims to determine the effectiveness of a complex transition care intervention delivered by nurses to medical patients planned to leave hospital care in reducing 30-day unplanned hospital readmissions or death. This important knowledge may utterly be considered for wider national and international dissemination as an evidence-based intervention for discharged patients at high risk for 30-day hospital readmission, identified through the simplified HOSPITAL score.

## Figures and Tables

**Figure 1 healthcare-11-00886-f001:**
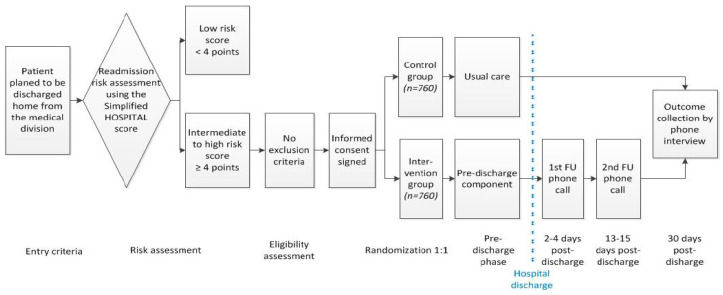
TARGET-READ study design.

**Figure 2 healthcare-11-00886-f002:**
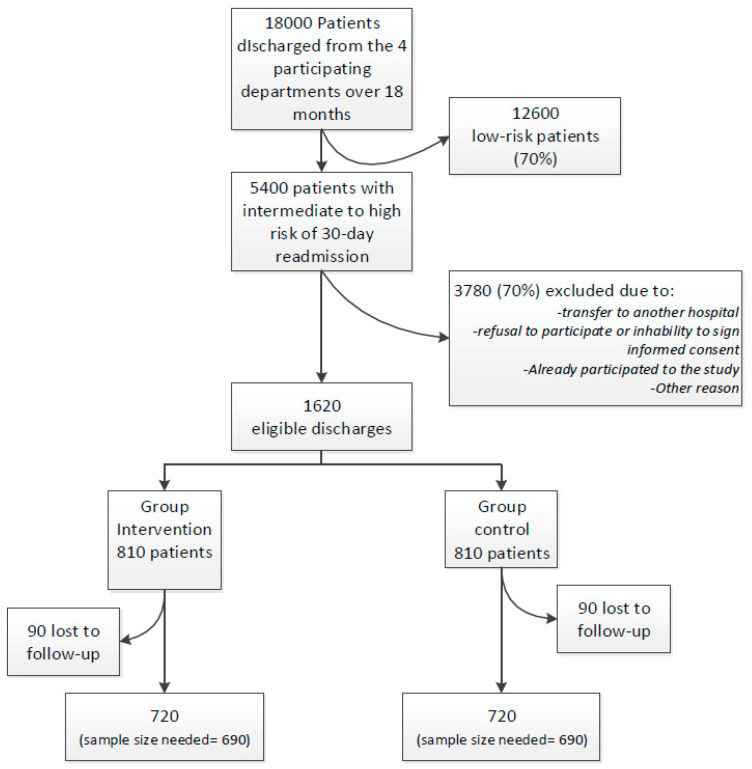
Projected study flow-chart with estimation of the number of included and excluded patients.

**Table 1 healthcare-11-00886-t001:** Simplified HOSPITAL score to identify the patients at high risk for 30-day readmission.

Attribute	Value	Points
Low hemoglobin level at discharge (<120 g/L)	Yes	1
Discharge from an oncology service or active cancer diagnosis	Yes	2
Low sodium level at discharge (<135 mmol/L)	Yes	1
Index admission type: urgent or emergent (nonelective)	Yes	1
Number of hospital admission(s) during the previous year	0–1	0
	2–5	2
	>5	5
Length of stay ≥ 8 days ^1^	Yes	2

^1^ Adapted and validated in Switzerland [24,25].

**Table 2 healthcare-11-00886-t002:** Inclusion and exclusion criteria.

Inclusion criteria	Adult patients planned to be discharged home or nursing home Discharge from a medical department of one of the four participating centers (Biel, Fribourg, Lausanne, Neuchâtel) Hospital stay of at least 24 h Patient at higher risk of 30-day readmission based on the simplified HOSPITAL score (≥6 points)
Exclusion criteria	Previous enrolment into the current study Not living in the country in the next 30 days No phone to be reached at Not speaking French or German (depending on the site) Refusal to participate, or unable to give consent (including cognitive troubles defined as a disorientation in time and space)

**Table 3 healthcare-11-00886-t003:** Study procedures and assessments pre and post-discharge.

	Index Hospitalisation ^1^		Follow-Up (Post-Discharge) ^2^		
	Screening (Before Inclusion	Visit 1 (Inclusion, Day 0)	Visit 2 (Day 3 ± 1)	Visit 3 (Day 14 ± 1)	Visit 4 (Day 30 ± 15)
Assessment	Pre-Study Screening	Baseline and Pre-Discharge Intervention	Post-Discharge Intervention 1	Post-Discharge Intervention 2	Outcome Collection
Demography	X				
Eligibility	X				
HOSPITAL score	X				
Informed consent		X			
Randomization		X			
Baseline characteristics		X			
Medical history		X			
Medication		X			
Activity of daily living (ADL Katz score)		X			
Exposition to intervention		X	X	X	
Medication discrepancy			X	X	
Adverse drug event			X	X	
Follow-up care plan followed			X	X	
Primary outcome: unplanned readmission or death					X
Secondary outcomes:					
Number of unplanned readmissions					X
Number of days of hospitalization within 30 days					X
Diagnoses at readmission or cause of death					X
Number of emergency department visits					X
Number of primary care physician visits					X
Discharge satisfaction (3-CTM)					X
Costs of readmission					X
Letter of discharge of the unplanned readmission					X
End of study					X

^1^ Assessment based on electronic health record and/or in person. ^2^ Assessment by phone call.

**Table 4 healthcare-11-00886-t004:** Documents transmitted in the intervention group (available as Appendix A).

Who	What
Each participant in the intervention arm	Information sheet about its participation to the study along with next appointment to the primary care physician General health hygiene information
According to the disease of the patient	One or two-page leaflet about frequent diseases
Primary care physician	Information sent directly to the primary care physician of the study participation of the patient and his/her high risk of being rehospitalized

## Data Availability

The datasets generated and/or analyzed during the current study are not publicly available due patient confidentiality but are available from the corresponding author on reasonable request.

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
