# Peer review of "Effectiveness of Transition Care Intervention Targeted to High-Risk Patients to Reduce Readmissions: Study Protocol for the TARGET-READ Multicenter Randomized-Controlled Trial"

_healthcare, 2023, doi:10.3390/healthcare11060886_

Round 1
Reviewer 1 Report
Dear Editor,
Thank you for the opportunity to read this study protocol entitled: Effectiveness of transition care intervention targeted to high-2 risk patients to reduce readmissions: study protocol for the 3 TARGET-READ multicenter randomized controlled trial.
I have only a few questions for the authors because the study is well-written and clearly presented.
First, please better define how is High-risk Patients, for example:
What is a high-risk patient?
What do we mean by risk?
Why do we want to assess risk?
How do we want to use this analysis?
Finally, I do not understand the setting.
Are preoperative patients excluded?
If yes, ok, but if not, what is the role of the perioperativist?
PMID: 27013123
Author Response
Thank you for the opportunity to read this study protocol entitled: Effectiveness of transition care intervention targeted to high-2 risk patients to reduce readmissions: study protocol for the 3 TARGET-READ multicenter randomized controlled trial.
I have only a few questions for the authors because the study is well-written and clearly presented.
Point 1:
First, please better define how is High-risk Patients, for example:
What is a high-risk patient?
What do we mean by risk?
Why do we want to assess risk?
How do we want to use this analysis?
Response 1: Many thanks for your comments and questions. Patients admitted in the recruting centers were stratified according to the risk of readmission through the simplified HOSPITAL score (Table 1). Patients with a total score of 6 or more were considered as high-risk patients. This information was added on lines 107, 124, 150 and on table 2, as a complement to the information that was previous available in line 175. The risk assessment is paramount for the identification of patients that are most likely to benefit from the intervention, as shown by previous research in the field.
Point 2:
Finally, I do not understand the setting.
Are preoperative patients excluded?
If yes, ok, but if not, what is the role of the perioperativist?
Response 2: Only patients admited in the internal medicine wards were considered as eligible, therefore preoperative patients were excluded (mentionned in line 114) and no perioperativist’ interventions were performed.

Reviewer 2 Report
In this article Gouveia et al. provided the study protocol for the TARGET-READ research project. The manuscript is well written and the appropriate research methodology followed. Moreover the authors provided a thorough statistical analysis plan. A minor comment to the authors:
-Please consider providing a more detailed description of the interventions in the experimental arm
-Additionally the authors could provide translated versions (as supplementary material) of the documents handled to the participants
Author Response
In this article Gouveia et al. provided the study protocol for the TARGET-READ research project. The manuscript is well written and the appropriate research methodology followed. Moreover the authors provided a thorough statistical analysis plan. A minor comment to the authors:
Point 1: Please consider providing a more detailed description of the interventions in the experimental arm.
Response 1: Many thanks for your questions and comments. Additional information was added to the manuscript with more details about the interventions in the experimental arm (lines 189 to 231).
Point 2: Additionally the authors could provide translated versions (as supplementary material) of the documents handled to the participants.
Response 2: Several supplementary files have been created and will be available as adittional material (as mentioned in lines 205 and 229).
